# Relationships Between Misinformation Variables and Nutritional Health Strategies: A Scoping Review

**DOI:** 10.3390/ijerph22060891

**Published:** 2025-06-03

**Authors:** Andrea Caballero, Cassandra Chapi-Nitcheu, Laura Vallan, Antoine Flahault, Jennifer Hasselgard-Rowe

**Affiliations:** 1Department of Global Health, Global Studies Institute, University of Geneva, 1211 Geneva, Switzerland; rea.caballero@etu.unige.ch (A.C.); cassandra.chapi-nitcheu@etu.unige.ch (C.C.-N.); 2Faculty of Medicine, Institute of Global Health, University of Geneva, 1211 Geneva, Switzerland; antoine.flahault@unige.ch (A.F.); jennifer.hasselgard-rowe@unige.ch (J.H.-R.)

**Keywords:** misinformation, nutrition, health campaign, public health

## Abstract

In an era where information is readily accessible, the dissemination of accurate and reliable health information is crucial for public health promotion. This scoping review explores the impacts of misinformation and disinformation variables on the implementation of nutritional health strategies. It also examines how the design and delivery of these strategies may contribute to the emergence of misinformation. By synthesizing insights from existing literature, this review highlights effective approaches and identifies research limitations to propose actionable recommendations for further investigation. A systematic search on PubMed, Embase, and Web of Science identified fourteen studies published between 2014 and 2024. These fourteen studies focused on various health strategies implemented across different media and in several formats to improve public knowledge and behavior regarding nutrition. Key findings included the role of misinformation variables in shaping public perceptions, the importance of cultural adaptation in health strategies, and the effectiveness of tools, such as e-health communication platforms. This review underscores the necessity for evidence-based, culturally sensitive, and accessible health promotion strategies to counter misinformation and foster informed nutritional practices. Further research is needed to address gaps in demographic representation, user engagement, and the long-term impacts of these strategies.

## 1. Introduction

In an era where information is readily accessible, the dissemination of reliable, accurate, and non-misleading health information is crucial for public health promotion. For this dissemination to occur, nutritional messages can be used to inform individual and community decisions that enhance health [1]. Such effective health communication can have a positive effect on knowledge, attitudes, and behaviors that promote well-being and prevent disease [2]. However, the proliferation of misinformation and disinformation poses significant challenges to these efforts [3]. Misinformation refers to false or misleading information shared without harmful intent, whereas disinformation involves deliberately deceptive content designed to mislead audiences [4]. Both phenomena can undermine trust in health authorities, distort public perceptions of health risks, and propagate ineffective or harmful practices [5]. In this review, we consider a range of variables associated with misinformation, including external variables (such as myths, fake news, and information quality) and internal variables (such as knowledge gaps, misconceptions, and low health or media literacy), without conflating them as equivalent or interchangeable.

In the digital age, health promotion strategies have embraced a variety of communication platforms, from traditional media to social media channels, to engage diverse populations. However, the proliferation of digital platforms has also fostered a parallel rise in misinformation and disinformation, posing significant challenges to public health initiatives [6]. It has been suggested that, instead of distributing evidence-based information that motivates real behavior change, social media often features dramatic content or idealized health outcomes to attract attention and gain followers [7].

More recent research on health and misinformation is related to the case of the COVID-19 pandemic, with a specific focus on the virus and vaccines. This research indicates that misinformation and disinformation can significantly affect public perceptions and behaviors [8]. Key publications in this area include studies by Vosoughi et al. [9], which highlighted the rapid spread of false news on social media, and by Lewandowsky et al. [10], which discussed the persistence of misinformation and strategies to correct it. Relevant to nutrition, Ruani and Reiss [5] identified nutrition misinformation linked to the prevention and treatment of COVID-19 on ‘lower-quality sources’, such as social media.

Nutritional health content, particularly surrounding diets, is commonly at risk for online misinformation [11]. Studies show that misconceptions about healthy eating are associated with a lower consumption of a healthy diet [3]. Media literacy, the ability to access and evaluate media content [12], is a promising approach to counter misconceptions about healthy diets and may support the mitigation of obesity risk in young people [13]. Another approach is to increase the transparency of nutritional content by introducing policy regulations on food labeling. Health claims on food packaging are often used as marketing strategies rather than to improve health and well-being [14]. Several researchers have shown that many of these health claims are misleading and therefore misinform consumers’ perceptions of a healthy diet [15,16].

The main aim of this review is to explore the relationships between variables associated with misinformation and nutritional health strategies. By synthesizing insights from existing literature, the review aims to highlight effective approaches and identify research barriers to propose actionable recommendations for further investigation [17]. The dissemination of existing research findings is essential for developing effective communication strategies that can counteract false or misleading information and promote healthy nutrition.

## 2. Methodology

### 2.1. Protocol

The scoping review was conducted in accordance with the Preferred Reporting Items for Systematic Reviews and Meta-Analyses—Extension for Scoping Reviews (PRISMA-ScR) statement [18]. No registration was created for this review.

### 2.2. Eligibility Criteria

To be considered eligible, articles had to include three aspects: nutrition, dis-/misinformation, and a health promotion strategy.

More specifically, what was broadly labeled as misinformation variables could involve, for instance, ‘misconceptions’, as in ‘being uninformed’. Nutrition could relate to food and beverage consumption (including nutrients, diet, and calories); mineral, probiotic, or supplement intake; and breastfeeding.

Health prevention strategies were defined as active, preventative campaigns or interventions, usually under a specific name and by an organization or team. Campaigns about the management or treatment of a condition instead of its prevention were excluded, as well as guidelines, as these were not considered active promotion. Many papers discussed plans for the creation of a campaign in light of already-established misinformation. Since the effects of ‘misinformation’ on the delivery of already-existing health promotion strategies were of interest, these were excluded. Several papers discussed misinformation and food labeling. Unless the specific food labeling approach was named and preventative, these articles were excluded as food labeling strategies, as they were either too broad to be considered a campaign or did not focus enough on health prevention. In the case of social media, studies that focused on a specific hashtag promoting nutritional advice were considered a health promotion strategy. Table 1 below presents an overview of the search criteria.

Peer-reviewed and gray literature published in English between 2014 and 2024 was included. Qualitative, quantitative, and mixed-methods studies were included. If they did not provide enough relevant detail for the search, reviews, meta-analyses, and conference proceedings were excluded.

### 2.3. Information Sources

In the literature search, the authors of this study ran searches on 3 engines: PubMed, Embase, and Web of Science, between October and November 2024.

### 2.4. Search Strategy

The search terms included the three concepts discussed in the research question: “misinformation”, “health strategy”, and “nutrition”. ChatGPT (November 2024 version) were used to identify variables associated with the scoping concepts. The terms were chosen and adapted by one author and checked and discussed with the other authors. As the literature did not always explicitly mention or distinguish between ‘misinformation’ and ‘disinformation’, we included search terms that could capture both definitions, such as ‘fake news’ and or ‘misleading information’. The search strings for Embase and Web of Science were as follows:

(misinform* OR disinform* OR misconcept* OR mislead* OR “false information” OR “fake news” OR myth* OR “misleading information”)

AND

(health AND prevent* AND (campaign* OR intervention* OR program* OR programme* OR initiative* OR “action plan” OR strategy*))

AND

(nutri* OR diet* OR eat* OR food OR “caloric intake”)

In PubMed, the search included MeSH terms for health promotion strategies (“Health Promotion” AND “Preventive Health Services”) and nutrition (“Diet, Food, and Nutrition”), as shown below:

(misinform* OR disinform* OR misconcept* OR mislead* OR “false information” OR “fake news” OR myth* OR “misleading information”) AND “Diet, Food, and Nutrition”[Mesh] AND (“Health Promotion”[Mesh] OR “Preventive Health Services”[Mesh])

### 2.5. Selection Process

Duplicates were removed after exporting the search results into Microsoft Excel. To ensure that all three authors were screening consistently using the same criteria, 10 records were first screened independently, marking each as Y, N, or M (‘yes’ = include, ‘no’ = exclude, and ‘maybe’ = to be screened fully and discussed by all members of the team). Another column was reserved for comments and justifications. Any differences were discussed until they were resolved unanimously. The authors drafted a criteria sheet (Table 1) to maintain consistency when screening the records.

The PubMed search results were screened independently by the authors. The search results from Embase and Web of Science were then divided equally among them.

As part of the full-text screening, the authors began charting any eligible reports and removed those that were (a) inaccessible; (b) not a health promotion strategy, nutrition, or misinformation, according to the set definitions; (c) reviews or meta-analyses that were too broad and did not provide enough relevant detail; or (d) not relevant to the research question.

### 2.6. Data Charting Process

At the end of the full-text screening, the reports marked ‘Y’ were charted. Excel was used for charting purposes; the table (see Table 2 and Appendix A) was developed jointly by all authors, based on the Preferred Reporting Items for Systematic Reviews and Meta-Analyses Extension for Scoping Reviews (PRISMA-ScR) checklist. The extraction included title, date, country/region, author(s), study aim, study design, population and sample size, intervention, source of misinformation, key findings, and shortcomings in the research. Interventions and sources of misinformation were aligned with the criteria in Table 1. Key findings were regarding misinformation.

### 2.7. Synthesis of Results

The data were summarized narratively in line with the data items. Additionally, health strategies were further summarized into intervention types: national or state-wide health strategies, multi-phase strategies, social media campaigns, and development of physical and e-health communication tools. Potential misinformation was categorized into three sources: information spread among specific populations, social media, and non-culturally adapted nutritional health strategies.

## 3. Results

Of the 948 records initially identified, 14 results aligned with our eligibility criteria mentioned in Section 2.2 and as such were included for analysis. The full data collection and screening process is presented in the PRISMA flowchart in Figure 1 below.

The characteristics of each study are shown in Table 2. For a full overview of the table, see Appendix A. Although the publication was limited to the last ten years, no relevant study was published before 2016. The most recent one was dated 2024.

### 3.1. The Studies’ Aims

The studies’ aims focused on designing effective and holistic public health strategies [23,26], counteracting nutritional fake news [19], enhancing nutritional labeling systems [20], promoting healthy behaviors through e-health communication tools [19], and assessing the effects of different health strategies [11,21,22,24,27,30]. Some studies explored how digital platforms and communication tools in healthcare settings influenced knowledge, attitudes, and behaviors related to nutrition, obesity prevention, and health promotion [25,28,29,31].

### 3.2. Study Designs and Methods

The study designs encompassed a diverse range of qualitative and quantitative methods. Bonvechhio Arenas et al. [23] described the planning of the intervention, the implementation, and the evaluation, while other studies assessed an intervention that was not part of the study itself [11,20,21,27]. The methods used by the study authors included qualitative approaches, such as focus groups, interviews, surveys, and observations. Two randomized controlled trials compared the outcomes of an intervention group to a control group [24,30]. One meta-study by Martínez-González et al. [22] was included due to its specific relevance to our study objective. Quantitative methods were not only used in the randomized controlled trials and the meta-analysis but also in a mixed-methods approach, mainly for evaluation via pre- and post-interventional surveys. Several studies were part of a larger-scale research project or intervention, on which other studies had previously been published [20,23,24,26,29,31].

### 3.3. Populations and Samples

Nine of fourteen studies included families as their target population [19,20,23,24,25,26,27,28,29]. Some studies were aimed especially at mothers [19,20,25,29], while others included caregivers in general [23,24,26,27,28]. Several studies focused on specific age groups, such as children up to 1 or 2 years of age [19,26,27], preschool children [24], school age [20], or college students [31]. Three of the health strategies that were in a family setting incorporated healthcare professionals’ expertise into their strategy development [20,23,26]. Low-income families, who generally faced more barriers to healthy eating behaviors, were only considered in a few studies [17,26,31].

The sample sizes included in the development and assessment of health strategies varied widely across the studies. Some included fewer than 50 participants [26,27,31], while others involved several hundred participants [24,25,28,29]. A few studies did not assess the effectiveness of a health strategy on an individual level [11,19,22], making the comparison across the studies more difficult. Bonvecchio Arenas et al. [23] included several hundred individuals in their research, strategy design, and feasibility study. Two health strategies used social media channels to reach around 9 million people each [19,21].

### 3.4. Geographical Locations

The studies were geographically distributed across several continents, with a notable focus on North America and Asia. Eight of the included studies investigated health strategies in the USA and Mexico [21,22,23,24,26,27,28,31]. Another four studies came from Asia, specifically from Iran [20], Nepal [29], India [27], and China [11]. Europe was represented by a study from Italy [19] and from Ireland [25]. Figure 2 presents an overview of the results by region. Some studies were on a community or city level [24,26,29,30,31], while others were on a larger scale. Three of the health strategies used exclusively online platforms and therefore reached across national and continental borders [11,19,21].

### 3.5. Health Strategies

Various health strategies were implemented to address the variables associated with misinformation in nutrition. These strategies spanned across different media and formats, aiming to improve public knowledge and behavior regarding nutrition. Prevention programs targeted various health issues, such as obesity [24,26,31], cardiovascular diseases [22,29], and kidney stones [21]. Dietary interventions included the promotion of gluten-free [11] and Mediterranean diets [22]. Formative research and needs assessments were explicitly mentioned in several studies to tailor the interventions effectively [19,23,26,27,28,30]. Workshops were conducted to design health strategies [23,26] or as part of their dissemination [29]. Several pilot projects were also mentioned, highlighting the initial testing and refinement of these strategies [23,28,30]. Cultural factors were considered in some health strategies, such as in the case of promoting the Mediterranean diet to non-Mediterranean populations [22] and addressing cultural aspects related to nutrition [23].

#### 3.5.1. National or State-Wide Health Strategies

National or state-wide health strategies included the implementation of traffic light labels and nutritional facts labels in Iran [20], the “Integrated Strategy for Attention to Nutrition” (EsIAN) in Mexico [23], and the “Babies Know the Facts about Folic” campaign by an Irish organization [25]. The “Cavities Get Around” campaign in Colorado aimed to raise awareness about children’s oral health. It involved stakeholders such as health promoters and community organizers who disseminated key messages to families and the community [28].

#### 3.5.2. Multi-Phase Interventions

Some strategies were implemented via multiple interventional time points. For instance, one study described two online activities that took place within three months and applied human-centered design principles [26]. The “Strategies for Effective Eating Development” (SEEDS) program included seven lessons over seven weeks, featuring videos and experiential learning activities for parents and children to promote healthy eating behaviors [24]. Another intervention included two nutritional education sessions of 40 min each within two weeks, aimed at improving daily fruit and vegetable intake [30]. The health strategy with the longest duration contained policy changes combined with monthly multi-component interventions over seven months [31]. Its activities included changes to food served, strategies to improve teachers’ health, classroom activities, training, and technical assistance. The “Heart-health Associated Research, Dissemination, and Intervention in the Community” (HARDIC) trial utilized peer education to conduct educational classes for mothers on children’s diet and physical activity behaviors [29]. Peer mothers were trained to conduct five health education classes within one month for groups of fellow mothers.

#### 3.5.3. Social Media Campaigns

Health strategies were implemented through hashtags on online video platforms such as TikTok and BiliBili [11,21]. These campaigns examined the reach, time trends, and quality of health content related to nutrition. Additionally, a social media marketing strategy was employed, whose details were not described in the study by Flaherty et al. [25]. One notable campaign, “Cavities Get Around”, utilized both traditional media, in particular, television and radio advertisements, along with social media platforms to disseminate information [28].

#### 3.5.4. Development of Physical and E-Health Communication Tools

Several physical visual campaign materials and communication tools were developed, including flipcharts, clinical manuals, scripts, visual aids, nutrition cards, pamphlets, PowerPoint presentations, training manuals, and posters [23,26,29,30]. E-health communication tools were also created to enhance knowledge and behavior regarding nutrition for mothers and children. These tools included a website, a social media page, and an app/chatbot [19]. Hammad and Kay [27] assessed the acceptance of a digital behavioral intervention tool designed to support the redemption of nutrient-rich food by members of the “Special Supplement Nutrition Program for Women, Infants, and Children” (WIC).

### 3.6. Sources of Potential Misinformation

The results showcased a range of variables associated with misinformation, whether external (such as myths, fake news, and information quality) or internal (such as knowledge gaps, misconceptions, and low health or media literacy), without treating them as synonymous. Though this review did search for disinformation, it was at times difficult to judge from the studies whether there was deliberate intent to distribute false information and therefore whether it could be classified as ‘disinformation’.

Broadly speaking, three sources of potential misinformation emerged as showin in Figure 3: (a) information spread among a specific population, such as pregnant women, parents, teachers, or college students [19,23,24,25,26,27,28,29,30,31], also the most common category; (b) social media platforms [11,21]; or (c) potential misinformation arising from non-culturally adapted health strategies [20,22].

These sources of potential misinformation interfered with the implementation or effectiveness of nutritional health strategies or shaped the aims of the strategy itself. The sources of potential misinformation were not always external to nutritional health strategies; at times, the sources were the strategies themselves. Seemingly, potential misinformation is not usually from one source but from a variety of sources that overlap and influence each other.

#### 3.6.1. Information Spread Among Specific Populations

Seven studies discussed pregnant women, parents, teachers, or college students as a source of potential misinformation regarding nutrition [24,25,26,28,29,30,31]. Meanwhile, three studies identified parents as receptors of potential misinformation and related variables in the context of child nutritional behaviors and went further to find the sources the parents relied on for nutritional information. Overall, these latter studies found that parents and caregivers could be influenced by online information, social media [27], peers [19], friends [27], family [23,27], primary healthcare providers or other health professionals [19,23,27], personal convictions [19,27], cultural beliefs [23], cooking shows [27], and research [27]. Each of these could be, but were not always, sources of potential misinformation and could result in a misinformed approach to nutrition. Hammad and Kay [27] demonstrated how the variety of sources of health information could lead to different definitions of ‘healthy’ versus ‘unhealthy’ eating. Another study highlighted how a lack of support for new parents, as well as shortcomings in their knowledge, could leave them vulnerable to inaccurate online information concerning infant nutrition [19]. However, while the influence of misinformation could lead to ‘unhealthy’ nutritional behaviors, there were other factors that played a role in these behaviors, such as a lack of economic resources and time [19].

#### 3.6.2. Social Media

Two studies explored the quality of nutritional information on social media, one concerning kidney stone disease on TikTok [21] and the other regarding the gluten-free diet on TikTok and BiliBili in China [11]. In the case of the online campaign for kidney stone disease, Salka et al. [21] found that while both physicians and non-physicians created the videos, most were from non-physicians, and the majority of the content did not align with the American Urological Association recommendations for diet therapies. In this context, TikTok served as a source of potential misinformation, given its unregulated and unverified health advice content. Similarly, the quality of the gluten-free diet videos on TikTok and BiliBili was found to be poor in terms of nutritional information, risking the dissemination of misleading, biased messages. Examples included videos promoting unproven treatments with no discussion of side effects [11].

#### 3.6.3. Non-Culturally Adapted Nutritional Health Strategies

Misinformation probability seemed greater in relation to a specific diet [22] and a labeling campaign [20] when transposed onto a new cultural context. In these new cultural contexts, the strategy itself could experience deviations that could be or lead to misconceptions and gaps in knowledge and indirectly lead to potential misinformation. For example, Martínez-González et al. [22] found that when the Mediterranean diet, or ‘MedDiet’, was adopted in other contexts, such as in the US, it experienced changes that did not align with the original dietary recommendations. These new, popular understandings of the MedDiet were informed by myths and misconceptions that interfered with the possibility of adopting the diet effectively in the USA [22].

In Iran, the traffic light labeling strategy (TLL) [20] was perceived as incompatible with the Iranian context. The strategy originated in contexts that are socio-culturally different, such as in the USA. In the study, various stakeholders, such as mothers, food quality control experts, nutritionists, and food industry experts identified the different issues in adopting the strategy with limited consideration of cultural differences. Mothers, given a lack of trust in manufacturers, were distrustful towards the labeling system. They sometimes perceived the information on packaging as unnecessary and at times did not have the necessary nutritional knowledge to understand the information. At the same time, food quality control experts believed the TLL to be misleading to consumers, particularly considering the lack of consistency in the definitions of the different categories. Additionally, the labeling strategies did not consider ingredients such as preservatives. As such, there were concerns that this could potentially result in manufacturers advertising reduced salt content by replacing salt with less healthy substances. Overall, one of the broader issues was a lack of regulation and supervision, which was different in the Iranian context than in the American one [20].

## 4. Discussion

### 4.1. Summary of Evidence

The studies revealed many components that could lead to potential misinformation in the population before the implementation of a health strategy or even within the nutritional campaign itself. The most important elements included difficulties in mobilizing information [19,20], poor adaptation and implementation of health strategies [20,22,27], and poor quality of information [11,21,22]. The findings also revealed the importance of addressing the burden of potential misinformation through campaigns [23,26] and the effectiveness of different health strategies in addressing this burden [24,25,28,29,30,31]. As demonstrated in the following subsections, the key findings frequently overlapped and may, in certain cases, complement one another. It is important to note that the findings should not be confused with misinformation in the strict sense but are rather inferences drawn from misinformation and related variables.

#### 4.1.1. Mobilization of Information

An important finding observed both externally and within nutritional strategies was the difficulty that populations faced when trying to mobilize information. The relationship between information overload and health misinformation started to gather interest since the COVID-19 pandemic, when there was a peak in the spread of health information through online platforms [32]. Oftentimes, the abundance of information available can make it difficult for individuals to distinguish between what is true and what is not. This can enable the spread of misinformation. The findings showed that the majority of pregnancy-related nutrition web pages contained some inaccurate information. An excess of available information made it difficult for parents to assess the quality of the information they were accessing [19]. In the study regarding the strengths and weaknesses of nutrition labels in Iran, a reported weakness from the mothers was the ambiguous and high amount of information provided [20]. The findings from these studies highlight that although it is important to have access to accurate information to tackle potential misinformation and related variables, it is just as important to ensure that the information is synthesized and presented in a way that target populations can understand without being overwhelmed.

#### 4.1.2. Poor Adaptation and Implementation of Strategies

While looking at the poor adaptation and implementation of nutrition strategies and potential misinformation, it is important to consider not only how poor adaptation can lead to potential misinformation but also how potential misinformation can lead to the poor adaptation of strategies. For example, in the implementation and promotion of a healthy diet, if misconceptions surrounding the diet are not properly addressed, this could lead to poor implementation of a nutritional strategy that would otherwise have proven health benefits [22]. Similar observations were found in a study where a few participants reported frustrations with the intervention because the information provided was sometimes seen as incomplete. For example, although the WIC program provided healthy foods, it did not provide guidelines to help cook or consume these foods. This highlights the importance of ensuring that interventions are targeted to the needs of the population [27]. The Iranian study on nutritional labels looked at both how distrust in information as a result of misinformation can lead to poor implementation and how poor implementation can lead to potential misinformation. This was shown through reported weaknesses by mothers, including unfamiliarity with the nutritional labels and culturally nonconforming labels. This was also supported by weaknesses reported by nutritionists and experts. The labels were at times considered misleading to consumers, or there was a perceived failure to properly implement the labeling strategy [20]. Policymakers and public health experts should therefore not neglect proper implementation and should assess and act upon common misconceptions that may surround their health strategy to avoid potential misinformation. Despite its importance, as highlighted by our review, further studies and investigations are needed on this specific aspect of misinformation in health strategies, as the available research is limited.

#### 4.1.3. Poor Quality of Information

The effects of poor-quality information in nutritional strategies were most visible in social media campaigns. Both social media campaigns reported low average DISCERN scores of the content, despite a very high reach [11,21]. This aligns with Wu et al.’s study [32] that found that large quantities of health information on social media are created and shared by ordinary users and not health professionals, which, in turn, spreads health misconceptions. Various studies consider how misconceptions are also present in strategies outside of social media, as can be seen in the study on the Mediterranean diet. In either case, this connects to the findings in the previous subsection on adaptation by showing how poor quality of information in a strategy can lead to potential misinformation, which can, in turn, continue the poor implementation of the strategy, as highlighted by Martínez-González et al. [22]. This underscores the need for future research to explore the interconnectedness of poor-quality information and poor implementation to address the burden of misinformation in health strategies.

#### 4.1.4. Importance and Effectiveness of Strategies in Addressing Potential Misinformation

Overall, the majority of the studies explicitly underscored the importance and effectiveness of nutritional campaigns and interventions in addressing misinformation. In one study from Mexico [23], local experts decided to develop and implement a strategy to tackle malnutrition with a focus on BCC—behavior change communication. In this case, BCC was a tool to help address the gaps caused by misconceptions. Although the findings of the study revealed that the strategy alone would not be enough to address the double burden of malnutrition and misconceptions, Bonvecchio Arenas et al. [23] demonstrated that it was still a good foundation. Similarly, in a study from Indianapolis, USA, after conversations with parents, researchers developed tools and a strategy to help address common misconceptions. They also found that such tools could help maximize the integration of prevention efforts across multiple settings [26]. Multiple studies also observed the effectiveness of interventions and nutritional strategies through measured changes in behaviors and an increase in knowledge [24,25,28,29,30]. These findings highlight how food literacy and nutritional education are effective tools in addressing and preventing misinformation that is caused by health illiteracy. For example, in the study by Patel et al. [30] in India, following a nutritional education intervention, correct knowledge regarding portion sizes and daily recommended number of servings of fruits and vegetables increased significantly from the baseline. Before the implementation of this intervention, the majority of the study population had less knowledge of this topic.

### 4.2. Limitations and Gaps in the Research

This review highlights various gaps and limitations in the research, including limited geographic and demographic representation, small sample sizes, and short-term study durations. These limitations put into evidence the necessity for more diverse, robust, and longitudinal studies on this topic. For example, the studies did not include South America and Africa, and therefore, the findings may not apply to these regions. Within the studies, there was also a lack of wider demographic representation [24,26,27,31]. Additionally, the studies frequently focused on child obesity and diet. Further studies could examine other aspects of nutrition.

Cultural beliefs significantly influenced eating behaviors. Although only two studies directly addressed cultural aspects in their health strategies [22,23], several others noted the absence of cultural considerations as a limitation [20,24,26]. Future research could not only adopt cultural frameworks to enhance the effectiveness of health interventions but could also explore the relationship between culture, potential misinformation, and nutrition.

A few of the studies had relatively small sample sizes [26,27,31], with the smallest one being n = 13. This can be viewed as a constraint, as too small of a sample may not always be an accurate representation of the population, and therefore, the results of these studies may not be generalizable. The majority of the studies were also relatively short-term and did not include multiple intervention stages. Several did not report any form of follow-up. This can be limiting, as it is uncertain if any reported change caused by the intervention was a permanent or temporary change [23,25,29,30,31]. Due to the nature of the strategies, a vast amount of data was self-reported by study participants. This can lead to desirability bias in certain cases and inflate the actual effect of the implementations [28,29,30,31].

When looking at larger-scale studies, such as the social media campaigns, the reach of these campaigns was discussed. However, there was scarce information on the interpretation or engagement of users in these situations. This is a caveat, as the effects of these campaigns—if any—cannot be fully assessed. For example, users may have encountered misinformation in these social media campaigns, but it remains unclear whether they believed the information or if they would act upon it [19,20,27]. Future research could explore the role of innovative digital tools and the promotion of media literacy to combat potential misinformation and related variables on a global scale.

The scoping review was based on a 10-year period that may have excluded prior studies on misinformation and related variables. As the search was conducted exclusively in English, this may have resulted in the regional emphasis on North America, limiting the inclusion of more geographically and culturally diverse studies. When setting the inclusion criteria, most studies on food labeling were excluded, as they did not identify a specific food labeling strategy or campaign. However, given the large quantity of these studies, further research could engage with potential misinformation among food labeling strategies. Another limitation included the lack of clear distinction between the terms ‘misinformation’ and ‘disinformation’. Although the search strategy aimed to capture variables related to both concepts, it was occasionally unclear whether the studies included in the analysis referred to false or misleading information disseminated intentionally or not.

## 5. Conclusions

This scoping review provides a comprehensive overview of how misinformation and related variables may influence health strategies around nutrition. The inferred findings indicate that misinformation and related variables, arising from various sources such as social media, cultural misconceptions, and non-culturally adapted health strategies, pose significant challenges to public health initiatives. Effective health promotion strategies must address these challenges by incorporating accurate, culturally relevant information and utilizing diverse communication platforms.

This review highlights the importance of tailored interventions that consider the specific needs and contexts of target populations. For instance, strategies that involve community engagement, such as peer education and human-centered design, show promise in enhancing the effectiveness of health promotion efforts. Additionally, the use of digital tools and social media campaigns can extend the reach of these interventions, although the quality and accuracy of the information disseminated through these channels must be carefully monitored.

Future research should prioritize broader geographic and cultural representation, longitudinal studies, and the integration of innovative digital tools to combat potential misinformation on a global scale. There is also a need to explore the role of media literacy in countering potential misinformation and to assess the long-term impact of health promotion strategies on behavior change. By addressing these areas, health strategies can become better tailored to combat potential misinformation and promote healthier nutritional behaviors.

## Figures and Tables

**Figure 1 ijerph-22-00891-f001:**
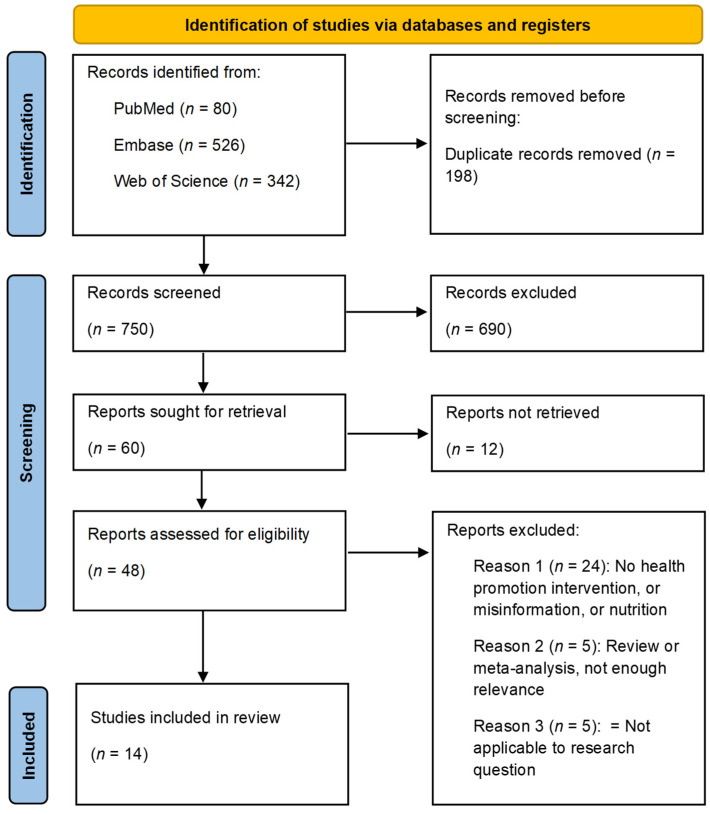
PRISMA flowchart.

**Figure 2 ijerph-22-00891-f002:**
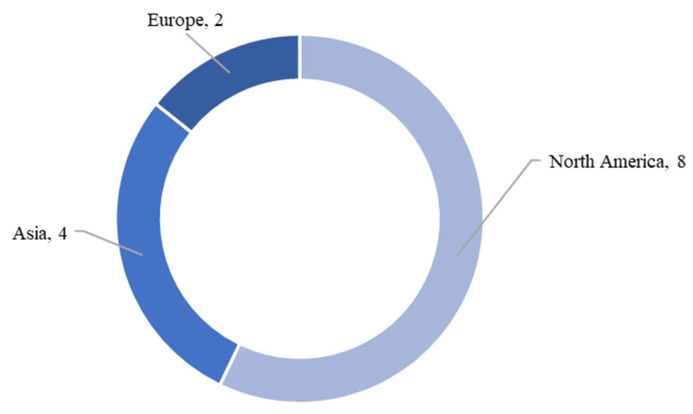
Number of included studies by region.

**Figure 3 ijerph-22-00891-f003:**
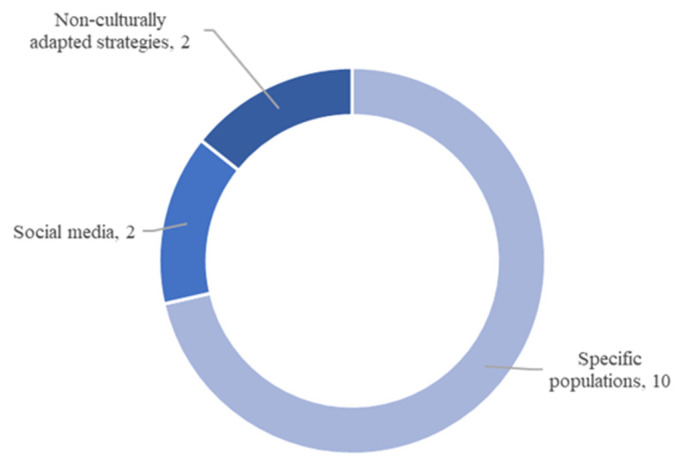
Sources of potential misinformation described in the results.

**Table 1 ijerph-22-00891-t001:** Criteria sheet: key research terms.

Term	Associated Terms	Exclusions
Misinformation	Disinformation; misinformation; misconceptions; lack of health literacy; unawareness; being uninformed	–
Health prevention strategies	Active promotion; defined and named campaign/intervention/strategy; social media hashtag campaigns that aimed to tackle misconceptions	Excluding campaigns that are management, not prevention; excluding guidelines—not active promotion; food labeling that was not prevention or that was discussed generally, e.g., on a national level
Nutrition	Supplements, probiotics, mineral intake, breastfeeding, nutrients, diet, calories	–

**Table 2 ijerph-22-00891-t002:** Characteristics of the included studies.

Title, Date, Country/Region	Author	Study Aim	Study Design	Population and Sample Size	Intervention	Source of Potential Misinformation
Nutripedia: The Fight against the Fake News in Nutrition during Pregnancy and Early Life, 2021, Italy (Online)	Verduci et al. [19]	Introduced novel tools of e-health communication with the scope of counteracting nutritional fake news concerning pregnancy	Interventional study using e-health communication tools (both qualitative and quantitative)	Parents Website engagement: 220,000 total views Social Media Reach: 9 million Chatbot: 14,698 downloads	Promoted new e-health communication tools, i.e., Nutripedia website and Chatbot app via non-governmental organization outreach, digital media sources, and scientific society conferences. Delivered general population advice, as well as individualized information and intervention. Information focused on nutritional knowledge (preconception, pregnancy, children up to 3 years).	Fake news concerning pregnancy and first 1000 days of life. Lack of support for new parents and obstacles in nutritional knowledge could leave them susceptible to inaccurate online information. Parents’ nutritional knowledge was affected by online tools, social environment, peers’ advice, medical assistance, and personal convictions.
Nutrition labels’ strengths & weaknesses and strategies for improving their use in Iran: A qualitative study, 2020, Iran	Seyedhamzeh et al. [20]	Explained the strengths and weaknesses of the traffic light label, and nutrition fact label and the strategies for improving their use in Iran	Qualitative Study—Focus Group Discussions	Mothers, *n* = 63; food quality control expert (FQC), *n* = 10; nutritionists, *n* = 6; and food industry experts, *n* = 8	Traffic light label (TLL) and nutrition facts label (NFL)	Mothers: Lack of trust in information provided by the manufacturers, incompatibility with the culture Otherwise, labels misleading the consumer, discrepancies in coloring reported by different laboratories, different approaches adopted by regulatory experts, ambiguity, and lack of supervision and consistency.
TikTok as an Educational Tool for Kidney Stone Prevention, 2023, USA	Salka et al. [21]	Evaluated the reach and quality of kidney stone prevention information on TikTok	Cross-sectional analysis	TikTok users Videos, English, related to hashtag/topic, with >1000 views, 87 videos, 8.75 million views	TikTok campaign #kidneystoneprevention	The majority of the TikTok videos, which did not meet American Urological Association recommendations for diet therapies in stone prevention.
Transferability of the Mediterranean diet to non-Mediterranean countries. What is and what is not the Mediterranean diet, 2017, USA	Martínez-González et al. [22]	Presented strategies necessary for promoting MedDiet to non-Med context, i.e., Americans	Cumulative meta-analysis	Those at risk for cardiovascular disease, 27 articles	Mediterranean Diet	Popular definitions of MedDiet were not in line with traditional diet, leading to myths/misconceptions and the promotion of foods that are not in line with MedDiet benefits.
Translating Evidence-Based Program Recommendations into Action: The Design, Testing, and Scaling Up of the Behavior Change Strategy EsIAN in Mexico, 2019, Mexico	Bonvecchio Arenas et al. [23]	Described the process and evidence-based approach used to design and rollout the EsIAN at scale. Focused on the behavior change communication component.	Mixed methods	Mothers/caregivers of children aged < 5 years, healthcare providers, experts, up to *n* = 1387, depending on the phase of the study	Integrated strategy for attention to nutrition, specifically the implementation of the behavior change communication (BCC) strategy component	Mothers: Misconceptions during pregnancy and breastfeeding, e.g., ‘they should eat for two during pregnancy’. Cultural beliefs could stem from mothers, mothers-in-law, fathers, and grandparents. Primary healthcare providers also sometimes lacked the knowledge concerning nutrition advice during pregnancy and lactation. Recommendations did not always align with global and national guidelines for mothers/caregivers.
Twelve-Month Efficacy of an Obesity Prevention Program Targeting Hispanic Families With Preschoolers From Low-Income Backgrounds, 2021, USA	Hughes et al. [24]	Assessed effects of an obesity prevention program which promoted eating self-regulation and healthy preferences	Randomized control trial	Hispanic preschool children (parents + families with low incomes) Families recruited from Head Start across 2 sites, 255 families randomized (prevention *n* = 136; control *n* = 119)	SEEDS (Strategies for Effective Eating Development) obesity prevention program. Curriculum aiming to change feeding knowledge/practices/styles (parent); BMI percentile, eating self-regulation, trying new foods, and fruit/vegetable consumption (child).	Mothers were sometimes a source of feeding misconceptions.
“Babies know the facts about folic”: A behavioral change campaign utilizing digital and social media, 2016, Ireland (Hybrid)	Flaherty et al. [25]	Assessed if the “Babies Know the Facts about Folic” campaign changed women’s knowledge, attitudes and behavior towards folic acid supplements	Online survey was conducted pre- and post-campaign. Home face-to-face interviews were conducted three months after the campaign.	Women of a childbearing age who were sexually active and could become pregnant online survey: *n* = 656/738, interviews: *n* = 424	“Safefood” launched a social and digital media campaign in 2015.	Misconceptions of women concerning the supplementation of folic acid prior to and during pregnancy could lead to low consumption of the supplement despite benefits.
Human-centered designed communication tools for obesity prevention in early life, 2023, USA (Indianapolis)	Cheng et al. [26]	Developed tools to ease the provider–parent communication about obesity prevention in a pediatric setting	Co-designed workshops with parents and pediatricians	Parents and caregivers of children aged 0–24 months, *n* = 13; and pediatricians, *n* = 13	Activities based on a human-centered design, involving parents and pediatricians. Activity 1: Discussion of communication barriers/facilitators in child obesity prevention. Activity 2: Creation of visual aids to facilitate the parent–provider communication.	There were knowledge barriers and misconceptions relating to infant feeding. In a needs assessment of parents of children aged 0–24 months, only a few parents viewed obesity in early life as a health issue.
Perspectives on healthy eating practices and acceptance of WIC-approved foods among parents of young children enrolled in WIC, 2023, USA	Hammad and Kay [27]	Analyzed what parents consider healthy food habits, their acceptance of certain food categories, and the acknowledgement of digital tools to improve diet quality	Qualitative interviews	Parents or caregivers of children aged 0–2 years who received benefits from the “Special Supplement Nutrition Program for Women, Infants, and Children”, had a cell phone, and spoke English, *n* = 13	The “Special Supplement Nutrition Program for Women, Infants, and Children” aimed to improve nutritional intake of low-income mothers and their 0–5-year-old children. Food packages containing vouchers for nutrient-rich foods were distributed to the intervention’s beneficiaries. However, the program has lately faced a decrease in participation, retention, and the redemption of food packages.	Mothers had different definitions of healthy versus unhealthy eating, possibly due to the variety of sources consulted to find nutrition information, e.g., social media, Google, YouTube, nutritionist from the intervention, friends, family, partners, cooking shows, research, and own common sense.
Busting the Baby Teeth Myth and Increasing Children’s Consumption of Tap Water: Building Public Will for Children’s Oral Health in Colorado, 2017, USA	Hornsby et al. [28]	Evaluated if a communication campaign could change behavior to limit children’s fruit juice consumption and increase tap water consumption to improve oral health	Pre- and post-campaign surveys applying quantitative methods	Low-income families living in Colorado who had a child between 6 months and 6 years of age, *n* = 603 (2014)/600 (2015)	“Cavities Get Around” is a statewide communication campaign including television and radio advertisements, social media, health promoters, educational programs, text messaging, and community partnerships to increase children’s consumption of tap water and decrease consumption of fruit juice and other sugary drinks.	Fruit juice was commonly regarded as healthy, although it contains a lot of sugar and therefore increases the risk for tooth decay in infants and children. Moreover, many parents underestimated the importance of baby teeth, regarding them as less important than adult teeth. Many parents did not know that cavities could spread from baby to adult teeth.
Effectiveness of health promotion regarding diet and physical activity among Nepalese mothers and their young children: The Heart-health Associated Research, Dissemination, and Intervention in the Community, 2019, Nepal (HARDIC) trial	Oli et al. [29]	Assessed the effectiveness of a health promotion intervention on mothers’ knowledge, attitudes, and practices, and their children’s behavior regarding diet and physical activity	Baseline and follow-up survey (quantitative methods)	Mothers of children aged 1–9 living in one of the two neighboring villages of Duwakot (intervention area) or Jhaukhel (control area), *n* = 323 mothers completed round 1 of the intervention *n* = 105 mothers completed round 2 of the intervention	The “Heart-health Associated Research, Dissemination, and Intervention in the Community”, (HARDIC) is a community-based health education program designed to improve diet and physical activity as part of cardiovascular health promotion. A total of 47 peer mothers were trained to conduct five education classes for about 10 mothers each.	There were widespread misconceptions by mothers who did not always understand the composition of healthy food.
Effectiveness of nutrition education in improving fruit and vegetable consumption among selected college students in urban Puducherry, South India. A pre-post intervention study, 2020, India (Puducherry)	Patel et al. [30]	Evaluated the effectiveness of nutrition education in improving the daily intake of fruit and vegetable servings and stage of behavior change among college students	Randomized control trial	Urban college students, *n* = 150	Intervention Group: 30 min nutrition education program Control Group: Pamphlets regarding healthy dietary intake	Initially, there was less knowledge regarding portion sizes, and average daily servings of fruit and vegetable intake among college students.
Gluten-free diet on video platforms: Retrospective infodemiology study, 2024, China	Ye et al. [11]	Examined the trends, content, and quality of information on two social media platforms	Mixed Methods: Mann–Kendall tests, DISCERN, HONcode	Videos using #GFD on TikTok and BiliBili, TikTok, *n* = 49 BiliBili, *n* = 86	Gluten-free diet videos	The quality of health information videos on Chinese video platforms were poor. The majority of them were not rigorous enough, containing misleading messaging that could promote unproven treatments with no discussion of possible side effects.
Influence of Teachers’ Personal Health Behaviors on Operationalizing Obesity Prevention Policy in Head Start Preschools: A Project of the Children’s Healthy Living Program (CHL), 2016, Hawaii	Esquivel et al. [31]	Quantified the Head Start (HS) teacher mediating and moderating influence on the effect of a wellness policy intervention	Intervention trial within a larger randomized community trial	Teachers from 23 Head Start (HS) classrooms, *n* = 46	Seven-month multi-component intervention with policy changes to food served and service style, initiatives for employee wellness, classroom activities for preschoolers promoting physical activity (PA) and healthy eating, and training and technical assistance	Knowledge, beliefs, priorities, and misconceptions around child nutrition among teachers could influence child nutrition and obesity.

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
