# Peer review of "Relationships Between Misinformation Variables and Nutritional Health Strategies: A Scoping Review"

_ijerph, 2025, doi:10.3390/ijerph22060891_

Round 1
Reviewer 1 Report
Comments and Suggestions for Authors
28-29 - This statement is incomplete: "the dissemination of accurate health information is crucial for public health promotion". Accurate information can still mislead. There are additional qualities besides accuracy that could be added in this statement - for example "non-misleading", "aligned with prevalent evidence", etc.
29-30 - This claim needs at least one citation, for example https://doi.org/10.1017/S0029665123000022
33 - It is misleading to imply there is only one definition of misinformation. It would be pertinent to include at least two definitions of misinformation by leading researchers and then choose the one that best supports the argument. Also, ensure the working definition captures elements of Table 1. At the moment, the authors' working definition of misinformation does not match what is implied in Table 1. Finally, be sure to reconcile the definitions or expand the conceptual framework with the terms listed in lines 280-284, as these currently present significant inconsistencies.).
36-37 - This claim needs at least one citation, for example https://doi.org/10.3390/nu15020451
42-44 - This sentence needs rephrasing (these words make no sense: "Instead of distributing evidence-based that..."). Elaborate on what is meant by "campaigns". Also, it is good practice to avoid stating observations as 'facts'- instead, use words like "it has been suggested that..."
45-51 - Needs to cover 'nutrition' misinformation in this context specifically, for example https://doi.org/10.3390/nu15020451
52-53 - This is misleading and needs rephrasing. Nutritional health content is not "a common category of online misinformation". Nutritional health content commonly carries misinformation risk, which is different from what is claimed here.
54-56 - Media literacy does not prevent obesity. This overstatement does not fully reflect what the citation says. Re-write.
69 - Using the word 'registration' is misleading, as this is not a registered trial and no registration of any kind is detailed.
78-79 - Fails to capture nutrients, diet, calories, eating, etc.
118 - Search terms are incomplete, partial, oversimplified, and don not reflect the promised scoping in the manuscript title. Ensure completeness.
218 - Graph lacks labels, %s and details to make it sufficiently informative.
279 - The manuscript fails to properly categorize all potential sources of nutrition misinformation already investigated in the literature, for example https://doi.org/10.3390/nu15214515
295 - Unreadable graph. Improperly labeled.
Substantial revisions needed:
1. The manuscript does not comprehensively capture existing research in diet and nutrition misinformation specifically, therefore it is not entirely reflective of the field of diet and nutrition misinformation. The studies included in the review appear to be more closely related to misconceptions, low nutrition literacy, low food literacy, and incompatibility with sociocultural views and attitudes, rather than misinformation in the strict sense, all of which are distinct concepts.
2. The conceptual framework needs to be reframed, as it conflates (confuses) misinformation with other connected concepts, such as misconceptions, low nutrition literacy, low food literacy, low knowledge/awareness, health attitudes, etc. These should not be confused as being the 'same' concept (even if they sometimes but not always overlap). Appropiate distinctions and categorizations need to be made.
3. Many of the citations do not correspond with the accompanying claims or assertions made in the manuscript, and others misrepresent the cited research.
4. The manuscript is also poorly cited - both in quantity and in quality, and requires pertinent citations to substantiate many of the claims and assertions made.
5. The PRISMA methodology is poorly applied and not properly elaborated. The manuscript misses many important steps of the PRISMA framework, such as Study risk of bias assessment, Effect measures, Reporting bias assessment, Certainty assessment. See https://prisma.shinyapps.io/checklist to help elaborate on the missing steps (this can be a mention of what was found/not found for each step, even if briefly - or why a step was skipped or not relevant).
6. The manuscript title does not reflect the findings and needs revising - for example, using the term 'disinformation' is misleading as the manuscript does not properly capture disinformation/manipulation techniques/terms.
7. The abstract is superfluous and lacks specificity or relevance to the intended research. It needs to be more informative and precise. Also, the research does not scope disinformation - for that, manipulation techniques and relevant search terms would need to be added to the methods. Unless the methods and searches are expanded, the term 'disinformation' should not be in the title, abstract, or front-page keywords.
8. The conceptual framework and methods need refining (substantially). With the appropriate revisions, this could be a useful manuscript, but at the moment it does not meet minimum required academic standards.
Comments on the Quality of English LanguageThe manuscript shows signs of superfluous phrasing, lack of specificity, and overuse of superficial terms like 'gap' (used 12 times). Consider more precise phrasing, ideally without reliance on generative AI (e.g. ChatGPT) for the writing.
Reviewer 2 Report
Comments and Suggestions for Authors
The work is extensive and has clearly looked at many studies, however, I feel that there is a disconnect between the title and what is in the main body of the manuscript. A nutritional health strategy would be something that one would expect an organisation to carry out. This may be the government, a large healthcare company or a large food company, I was therefore a little confused when reading this as to what the aim of the study was. I believe what is meant, is the individual person's health strategy. I think that it would be clearer if the title were to state 'The influence of misinformation on public perception of a healthy diet' or something to this effect. To me, this is what your work has looked at. This would also involve making the results a little clearer, by reflecting more on how the findings of the studies impact the individual, you have done most of this already. In places it would be good to name the authors of the papers, to which you refer, rather than just keep listing the reference list number, as this would make it easier for the reader to relate to, especially in the results section. I hope that these comments help you focus on exactly what you are trying to say in your paper, as at present it is a little hard to align the aims to what you have found.
I do believe that this can be done quite easily and with a bit more clarity this will make a good paper.
Other specific comments:-
Line 42 should read evidence base not evidence based.
Line 56 - where you talk about food labelling and policy regulation, this really only affects the individual, as highlighted in my earlier comments.
Line 88 - the same can be said for hashtags, these would not affect health strategies, it would be the hashtag (which may contain a health policy) that would affect an individual.
Line 129/130 you have repeated the sentence.
Round 2
Reviewer 1 Report
Comments and Suggestions for Authors
Please refer to the comments provided in the attached file.

Please refer to the comments provided in the attached file.
Round 3
Reviewer 1 Report
Comments and Suggestions for Authors
The revisions have improved the academic standing of the work.